# TFAM Loss Induces Oxidative Stress and Divergent Phenotypes in Glioblastoma Metabolic Subtypes

**DOI:** 10.3390/ijms262110446

**Published:** 2025-10-27

**Authors:** Stella G. Cavalcante, Roseli da S. Soares, Miyuki Uno, Maria J. F. Alves, Ricardo C. Cintra, Paula R. Sola, Christiane Y. Ozaki, Antonio M. Lerário, Sueli M. Oba-Shinjo, Suely K. N. Marie

**Affiliations:** 1Cellular and Molecular Biology Laboratory (LIM 15), Department of Neurology, Faculdade de Medicina FMUSP, Universidade de São Paulo, Room 4110—Pacaembu, 4th Floor, Av. Dr. Arnaldo, 455, São Paulo 01246-903, SP, Brazil; stellagcavalcante@usp.br (S.G.C.);; 2Centro de Investigação Translacional em Oncologia—CTO, Instituto do Câncer do Estado de São Paulo—ICESP, São Paulo 01246-000, SP, Brazil; 3Department of Radiology and Oncology, Faculdade de Medicina FMUSP, Universidade de São Paulo, São Paulo 05403-010, SP, Brazil; 4Department of Internal Medicine, Division of Metabolism, Endocrinology and Diabetes, University of Michigan, Ann Arbor, MI 48109, USA

**Keywords:** TFAM, mitochondrial dysfunction, GBM, redox balance

## Abstract

Mitochondrial transcription factor A (TFAM) is essential for mitochondrial DNA (mtDNA) maintenance and function, but its role in glioblastoma (GBM) remains largely unexplored. Analysis of patient astrocytomas and TCGA datasets has revealed progressive *TFAM* downregulation with increasing malignancy, with the lowest expression in glycolytic/plurimetabolic (GPM) subtypes. Functional and transcriptomic profiling of mesenchymal GBM cell lines showed that TFAM silencing in GPM-type U87MG cells enhanced proliferation, S-phase entry, reactive oxygen species (ROS) production, and adhesion, while reducing motility. These changes were correlated with upregulation of *LDHC* and *TRAF2* and downregulation of androgen receptor-linked motility genes and *LOXL2*. By contrast, TFAM loss in mitochondrial (MTC)-type A172 cells caused minimal phenotypic alterations, associated with elevated *SOD1* expression and activation of antioxidant, mitochondrial membrane, and survival pathways, alongside suppression of oxidative phosphorylation and vesicle-trafficking genes. TFAM overexpression reduced proliferation in U87MG but had a limited impact on A172 cells. Taken together, these findings establish TFAM as a subtype-specific regulator of GBM cell proliferation, redox balance, and motility. TFAM loss drives a proliferative, ROS-sensitive phenotype in GPM-type cells, while eliciting adaptive, stress-resilient programs in MTC-type cells. This study identifies TFAM and downstream effectors, *TRAF2* and *LOXL2*, as potential therapeutic targets, supporting the development of metabolic subtype-tailored strategies for GBM treatment.

## 1. Introduction

Mitochondria are widely recognized as the energy centers of eukaryotic cells, residing in the cytoplasm and primarily responsible for generating adenosine triphosphate (ATP), the molecule that fuels a wide range of cellular functions. ATP production can occur via glycolysis, a cytosolic pathway that converts macronutrients, such as glucose or fructose, into pyruvate. For complete oxidation, pyruvate is transported into the mitochondrial matrix, where it undergoes the citric acid cycle (TCA) and oxidative phosphorylation (OXPHOS), coupled with the electron transport chain—processes that account for the majority of cellular energy yield [1,2]. Uniquely, mitochondria are the only organelles in cells to have their own genetic material, mitochondrial DNA (mtDNA), located in the mitochondrial matrix and packaged by mitochondrial transcription factor A (TFAM) that binds to the single transcription promoter of mtDNA and regulates folding and opening [3]. The packaging of mtDNA by TFAM forms a nucleoid, protecting the mtDNA from environmental vulnerability to genotoxic damage from reactive oxygen species (ROS) produced by the respiratory chain [4,5]. TFAM is encoded by the nuclear gene of the same name, and its transcription is regulated by three main transcription factors: nuclear respiratory factor 1 (NRF-1), PGC-1α (co-activator-1 ‘alpha’ of the peroxisome proliferator-activated receptor), and nuclear respiratory factor 2 (NRF-2). Indeed, TFAM can remain in the cell nucleus and self-regulate its expression through protein–protein interactions inhibiting NRF-1, generating negative feedback as a result of high metabolic activity or amount of mtDNA [6,7,8,9,10,11].

Cancer is characterized by the uncontrolled growth of cells, driven primarily by the activation of oncogenes and the inactivation of tumor suppressor genes. These genetic alterations promote genomic instability and foster tumor development through hallmark processes such as unchecked proliferative capacity, resistance to cell death, replicative immortality, induction of angiogenesis, immune evasion, and deregulated cellular bioenergetics [12]. Gliomas are the most prevalent type of brain tumor, accounting for more than 80% of cases, and rank among the most lethal forms of cancer. Gliomas arise from glial cells and are classified by malignancy grade into diffuse astrocytomas (Grade 2-AG2), anaplastic astrocytomas (Grade 3-AG3), and glioblastomas (Grade 4-GBM). GBM represents the most aggressive form, marked by profound intratumoral heterogeneity, diffuse invasiveness, and a high rate of recurrence [13]. Despite the current standard of care—which includes maximal surgical resection followed by radiotherapy and chemotherapy with temozolamide (TMZ)—patients with GBM face a dismal prognosis, with a median overall survival of ~15 months only [14]. GBMs can be stratified according to their molecular profiles into three major subtypes: proneural (PN), classic (CS), and mesenchymal (MS)—the latter type being associated with the poorest prognosis [14,15]. More recently, an alternative classification has been proposed based on the predominant signaling pathways activated within these tumors. This framework defines four subgroups: proliferative/progenitor (PPR), neuronal (NEU), mitochondrial (MTC), and glycolytic/plurimetabolic (GPM). The GPM subgroup exhibits high molecular similarity to the mesenchymal subtype and is likewise associated with unfavorable clinical outcomes. Interestingly, the GPM subtype was associated with high metabolic activity besides mitochondrial pathways, which, in contrast, is predominant in the MTC subtype [16]. This refined stratification highlights the central role of metabolic reprogramming in gliomagenesis, underscoring its importance beyond the classical Warburg effect, a hallmark feature of cancer [12,17]. Our group has previously reported that *TFAM* expression decreases with increasing tumor grade. We previously reported that *TFAM* expression declines with increasing tumor grade, with low-grade astrocytomas (AG2 and AG3) exhibiting higher levels than GBM (AG4). Elevated *TFAM* expression correlated with improved patient prognosis, suggesting a protective role in glioma progression [18]. These findings underscore the close interplay between metabolic reprogramming and malignancy, yet the signaling pathways underlying TFAM dysfunction remain largely undefined.

## 2. Results

### 2.1. Subsection TFAM Expression Analysis in Human Astrocytomas Shows Significantly Higher Expression in Lower-Grade Astrocytomas

Consistent with previous findings of our group [18], *TFAM* expression progressively decreased with increasing tumor grade, from AG2 to GBM. In the cohort assessed, *TFAM* levels were significantly lower in GBM relative to both AG2 (*p* < 0.0001) and AG3 (*p* = 0.0089; one-way ANOVA, Kruskal–Wallis test with Dunn’s post hoc test) (Figure 1A). This trend was confirmed in The Cancer Genome Atlas (TCGA) dataset, comprising 363 astrocytomas (65 AG2, 131 AG3, and 167 GBM), where *TFAM* expression was again significantly higher in both AG2 (*p* < 0.0001) and AG3 (*p* = 0.0006) relative to GBM (Figure 1B).

Stratification of TCGA-GBM samples according to Garofano’s metabolic classification [16] revealed that the GPM subtype (n = 34) exhibited significantly lower *TFAM* expression than the MTC subtype (n = 43; *p* = 0.0003, Mann–Whitney test) (Figure 1B). Likewise, *TFAM* expression in the GPM subtype was lower than in the NEU and PPR subtypes (Appendix A). Notably, the GPM subtype—characterized by low *TFAM* expression—has been associated with the poorest prognosis among GBM patients.

At the protein level, TFAM immunostaining was assessed in 79 astrocytoma samples, comparing low-grade gliomas (AG2 and AG3) and GBMs (Figure 1C). Consistent with transcriptomic findings, TFAM protein levels were significantly lower in GBM than in AG2 (*p* = 0.0151) and AG3 (*p* < 0.0001; one-way ANOVA, Kruskal–Wallis test with Dunn’s post hoc test). These samples were not stratified further by metabolic subtype.

Two MS GBM cell lines with distinct metabolic profiles, previously characterized by our group [19], were selected to investigate the role of TFAM in phenotype determination. U87MG cells exhibit GPM features, whereas A172 cells display MTC characteristics.

### 2.2. TFAM Transient Silencing Characterization

A significant reduction in *TFAM* mRNA expression was observed in both cell lines following siRNA-mediated silencing (siTFAM), compared with non-targeting control (NTC) cells, beginning at day 2 and persisting through day 7. In U87MG-siTFAM cells, *TFAM* mRNA levels decreased by 11%, 23%, and 25% at days 2, 4, and 7, respectively (*p* = 0.0032, *p* = 0.0062, *p* = 0.0059, respectively; two-way ANOVA, with Bonferroni post-test). In A172-siTFAM cells, reductions of 5%, 12%, and 18% were observed at the same time points. Notably, mtDNA copy number declined in parallel with *TFAM* expression, remaining reduced until day 7, with decreases of 53%, 17%, and 15% (*p* = 0.0124, *p* = 0.0014, *p* = 0.0015, respectively; two-way ANOVA, with Bonferroni post-test) in U87MG-siTFAM, and of 66%, 25%, and 20% in A172-siTFAM, at days 2, 4, and 7, respectively (*p* = 0.0439, *p* = 0.0024, *p* = 0.0018, respectively; two-way ANOVA, with Bonferroni post-test) (Figure 2A,B). Consistent with these findings, TFAM protein levels also declined from day 2 onwards (in U87MG D2: *p* = 0.0008, D4: *p* = 0.0002, D7: *p* = 0.0003; and in A172 D2: *p* = 0.0042, D4: *p* = 0.0007, D7: *p* = 0.0005; two-way ANOVA, with Bonferroni post-test), reaching the lowest expression at day 4 for both cell lines (15 ± 8.18% in U87MG-siTFAM and 22 ± 4.74% in A172-siTFAM) (Figure 2C,D). Based on these results, subsequent functional and transcriptomic analyses were performed at day 4 of TFAM silencing, corresponding to the point of maximal downregulation at both the transcript and protein levels.

### 2.3. U87MG-siTFAM Cells Exhibit Enhanced Proliferation and Increased S-Phase Entry

U87MG-siTFAM cells showed a significantly higher proliferation rate compared with NTC controls at both days 4 and 7 post-transfection (*p* < 0.0001 at both time points; two-way ANOVA with Bonferroni post-test) (Figure 2E). Consistent with this rise, cell cycle analysis revealed a significant increase in the S-phase population (8.75 ± 0.98%, *p* = 0.0029) (Figure 2F).

By contrast, A172-siTFAM cells showed a transient increase in proliferation at day 4 post-transfection (*p* = 0.0004; two-way ANOVA with Bonferroni post-test), which was not sustained at day 7 (Figure 2G). No significant changes were observed in the cell cycle distribution of A172-siTFAM cells (Figure 2H).

### 2.4. U87MG-siTFAM Cells Exhibit Impaired Motility and Increased Adhesion

TFAM silencing significantly reduced the migratory capacity of U87MG cells, as demonstrated by delayed wound closure at 12 and 24 h post-wounding (*p* = 0.0286 and *p* = 0.0058, respectively) (Figure 3A,B) and by reduced transwell migration in the Boyden’s chamber assay (*p* = 0.0002) (Figure 3D,E). In addition, U87MG-siTFAM cells showed increased substrate adhesion in a short-term adhesion assay (3 h) compared with NTC controls (*p* = 0.0002) (Figure 3C).

Conversely, A172-siTFAM cells displayed no significant changes in migratory behavior in either the wound healing or Boyden’s chamber assays (Figure 3F–J). Similarly, no differences in adhesion were detected between A172-siTFAM and NTC cells (Figure 3H).

### 2.5. U87MG-siTFAM and A172-siTFAM Display Differential Responses to Oxidative Stress

Given the central role of TFAM in mitochondrial function, oxidative stress was assessed by quantifying ROS following TFAM silencing, using flow cytometry-based assays for both total and mitochondrial ROS. In U87MG-siTFAM cells, intracellular ROS levels, measured by H_2_DCFDA conversion, were significantly elevated compared to NTC controls (1.78× increase; *p* = 0.0325, two-way ANOVA with Bonferroni’s post-test), whereas no significant change was detected in A172-siTFAM cells (0.98× increase, *p* = 0.2956) (Figure 4A). Similarly, mitochondrial superoxide production, measured by MitoSOX, was significantly increased in U87MG-siTFAM cells relative to NTC controls (1.60× increase, *p* = 0.0005; two-way ANOVA with Bonferroni’s post-test), but remained unchanged in A172-siTFAM cells (Figure 4B).

In a previous study by our group [19], the expression of genes involved in the antioxidant defense pathway in both U87MG and A172 cell lines was analyzed. Notably, basal expression patterns observed of *SOD1* and *SOD2* differed between the two cell lines, with A172 cells exhibiting higher *SOD1* levels than U87MG cells. (Figure 4C). This disparity suggests that U87MG cells possess a comparatively weaker antioxidant buffering capacity, consistent with the elevated ROS accumulation observed in U87MG-siTFAM cells.

### 2.6. Transcriptomic Profiles of U87MG and A172-siTFAM Cells

To elucidate the signaling pathways affected by TFAM downregulation, transcriptomic analyses were performed on siTFAM and NTC cells.

In U87MG-siTFAM cells, 2990 genes were differentially expressed (DEGs) out of 18,793 mapped, including 1327 significantly upregulated (adjusted *p*-value AdjP ≤ 0.05). Gene ontology (GO) enrichment of these upregulated genes (FDR < 0.003) revealed a strong association with cell cycle regulation, particularly chromosome segregation, regulation of chromosome organization, positive regulation of cell cycle, chromosome condensation, and organelle fission—highlighting their role in mitotic progression. Conversely, 1663 genes were significantly downregulated, with the top enriched pathways linked to epithelial cell proliferation, angiogenesis, blood vessel development, and cell motility (e.g., regulation of cell migration and of cellular component movement) (Figure 5A).

Network analysis revealed complex interactions between these DEGs (Figure 5C). TFAM silencing disrupted mitochondrial biogenesis through downregulation of *PPARGC1A* (PGC1α) and its interactor, androgen receptor (*AR*; logFC = −2.2). *AR* connectivity extended to key cell cycle regulators, including *ELMO1*, *HRAS*, *CCNA1*/*2*, *MAPK1*/*9*, *BORA*, and *AURKA*, all of which were upregulated. Similarly, *TUBB4A*, associated with mitosis, was increased. On the other hand, *AR* was linked to *KIT*, a central regulator of cell motility, which in turn connected with integrins and multiple extracellular matrix (ECM)-related genes such as *COL4A1*, *COL18A1*, *LAMC3*, and *SDC2*. Several motility-and angiogenesis-related genes (*SLIT2*, *LIMK1*, *UNC5C*, *FZD9*, and *TIAM2*) were downregulated, three of which (*UNC5C*, *FZD9*, and *TIAM2*) also play roles in angiogenesis. Notably, TFAM silencing markedly upregulated *LDHC* (logFC = 6.4), while *LDHA* and *LDHB* remained unchanged, suggesting a shift toward an alternative non-oxidative metabolic pathway.

In A172-siTFAM cells, 3239 DEGs were identified out of 18,793 mapped, including 1636 that were significantly upregulated. GO enrichment analysis of these upregulated genes showed pathways related to mitochondrial regulation, including mitochondrial outer membrane permeabilization and regulation of mitochondrial membrane permeability during apoptosis. The GO term “lysosome localization” was also enriched, suggesting a potential link to oxidative stress. Additionally, the cell cycle-associated term “mitotic metaphase plate congression” was enriched, reflecting ongoing mitotic activity (Figure 5B). Among 1603 downregulated DEGs, enriched pathways included mitochondrion organization, energy derivation from oxidation of organic compounds, macroautophagy, and organelle localization.

Network analysis highlighted interconnected pathways (Figure 5D). Several OXPHOS genes were downregulated, including components of complex I (*NDUFA4*, *NDUFA6*), complex IV (*COX7B*), ATP synthase (*ATP5V1A*), and oxidative metabolism regulators (*MTOR*, *MTTL14*). Downregulation of *VAMP7*, *ANK1*, *CAV2*, *ATG14*, and *GCGR* (logFC = −1.7), vesicle trafficking-related genes, suggested impaired intracellular transport and metabolic flux. In contrast, genes regulating mitochondrial membrane dynamics were upregulated and connected with mitotic regulators such as *E2F1* and *CCND1*. Importantly, *GSK3A* and genes coding 14-3-3 proteins (*YWHAG*, *YWHAE*, and *YWHAQ*) emerged as central hubs bridging mitotic progression and oxidative stress pathways.

Furthermore, survival-promoting genes—including *BAK1*, *TNFAIP3*, *PPIF*, *HSPB1*, *RELA*, *TLR4*, and *ILB* (logFC = 2.5)—formed coordinated networks linking metabolic processes, mitochondrial membrane regulation, and mitotic activity. Taken together, these findings suggest that TFAM-deficient A172 cells activate an integrated molecular program aimed at preserving mitochondrial integrity, mitigating oxidative stress, and sustaining proliferation, ultimately balancing mitotic progression with apoptotic signaling.

### 2.7. TFAM Overexpression

To determine whether the effects observed upon TFAM silencing can be reversed, TFAM was overexpressed in both cell lines. Two multiplicities of infection (MOIs) were tested, and, for each cell line, the MOI that yielded the greatest increase in *TFAM* mRNA expression and mtDNA copy number was selected.

In U87MG cells, both overexpression constructs (TF1c1 and TF2c1) produced a significant increase in *TFAM* transcript levels compared with empty vector control (Vɸ), with statistical significance for U87MG-TF1c1 (*p* = 0.0031) and U87MG-TF2c1 (*p* = 0.0003, one-way ANOVA with Tukey’s post hoc test) (Figure 6A, left panel). Consistent with these results, mtDNA copy number was significantly elevated in both conditions relative to Vɸ (U87MG-TF1c1: *p* = 0.0288; U87MG-TF2c1: *p* = 0.0385) (Figure 6A, right panel). Western blotting confirmed increased TFAM protein expression for both constructs (TF2c1: 193.29 ± 15.47%, *p* < 0.0001), with a particularly pronounced upregulation in U87MG-TF1c1 (201.40 ± 10.14%, *p* < 0.0001) (Figure 6B).

Cell proliferation assays revealed a significant reduction in cell number at 48 h (*p* = 0.0359) and an even stronger effect at 72 h compared to Vɸ controls for both constructs (*p* < 0.0001, two-way ANOVA with Bonferroni’s post-test) (Figure 6C).

By contrast, the A172 cell line exhibited inefficient overexpression at both MOIs tested, resulting in a complete loss of viable cells and precluding further functional analysis.

### 2.8. TFAM Levels Predict Cell Cycle and Motility Pathway Activity and Identify TRAF2 as a Potential Therapeutic Target in GPM-GBM

To evaluate the clinical relevance of TFAM-associated pathways, genes previously identified as DEGs in U87MG-siTFAM cells, particularly those linked to cell cycle and cell motility, were analyzed in the TCGA GBM dataset. Among the 77 GBM cases, 62% were of the MS molecular subtype (n = 48), and within these, the highest percentage of GPM cases was observed (45%, n = 22 cases). Due to the alterations presented in this study, the analyses continued in the MS-GBMs. The 33 GPM and MTC cases from MS-GBM were stratified as “*TFAM* upregulated” or “*TFAM* downregulated” according to a cutoff determined by the receiver operating characteristic (ROC) analysis. A heatmap of the genes significantly associated with *TFAM* expression (Welch’s test, *p* ≤ 0.05), ordered by *TFAM* levels and clustered by pathway, revealed three major gene clusters in GPM tumors: (i) cell cycle-related upregulated genes; (ii) cell motility-related downregulated genes; and (iii) cell motility-related upregulated genes (Figure 7A). Among the genes showing significant differences in GPM versus MTC subtypes, the cell cycle regulator *TRAF2* was notably downregulated in the *TFAM*-upregulated group (*p* = 0.0279, Mann–Whitney test) (Figure 7B) and exhibited a negative correlation with *TFAM* expression (r = −0.37, *p* = 0.0009, Spearman correlation test) (Figure 7C). Kaplan–Meier (KM) survival analysis demonstrated that low *TRAF2* expression was significantly associated with improved overall survival (*p* = 0.041, log-rank test) (Figure 7E). Although the combined status of *TFAM* upregulation and *TRAF2* downregulation showed a trend toward improved survival, this did not reach statistical significance, likely due to limited sample size (Figure 7F).

Given the enrichment of motility-associated pathways, the relationship between *AR* signaling and cell motility genes was subsequently investigated. Among these genes, *LOXL2* was significantly upregulated in the *AR*-downregulated group (*p* = 0.0077, Mann–Whitney test) (Figure 7B) and showed a negative correlation with *AR* expression (r = −0.36, *p* = 0.0360, Spearman test) (Figure 7C). Survival analysis revealed that *LOXL2* downregulation was associated with significantly improved survival (*p* = 0.048, log-rank test) (Figure 7E). Although the combined analysis of *LOXL2* downregulation with *AR* upregulation suggested a trend toward longer survival, this also did not reach statistical significance (Figure 7E). Similarly, analysis of the combined status of *LOXL2* downregulation and *AR* downregulation—reflecting general suppression of motility-related pathways—was precluded by sample size limitations.

## 3. Discussion

Mitochondria are central regulators of cellular metabolism and homeostasis, and their function is frequently altered in tumor cells due to mutations in key driver genes, including those involved in the TCA cycle. TFAM, a master regulator of mtDNA transcription and replication, plays a critical role in mitochondrial biogenesis and metabolic regulation. Altered TFAM expression has been reported in several tumor types. Consistent with this, our group previously demonstrated a progressive decrease in *TFAM* mRNA levels with increasing astrocytoma malignancy, reaching the lowest levels in GBM, particularly in the MS and GPM subtypes—those associated with the worst prognosis [18]. Reduced *TFAM* expression has also been linked to tumor progression in colorectal cancer [20], chemoresistance in ovarian carcinoma [21], invasion in melanoma [22], and metastatic potential in pancreatic ductal adenocarcinoma [23].

At the molecular level, TFAM not only governs mtDNA transcription and replication [14,22] but also influences nuclear gene expression, including repression of its own transcription by inhibiting NRF-1 [9] and regulating *BIRC5* and *CDKN1A*, both of which contribute to tumorigenesis [24]. Reduced TFAM levels impair nucleoid stability, compromise OXPHOS activity, and disrupt cellular fitness [25,26,27], as reported in renal tubular cells and fibroblasts [28,29]. Thus, decreased TFAM is generally expected to attenuate cell growth.

Surprisingly, our data show that TFAM silencing in U87MG GBM cells increased cell viability, consistent with reports in esophageal squamous cell carcinoma [30] and head and neck cancers [31]. Conversely, TFAM downregulation in gastric [32] and hepatocellular carcinomas [33] reduced proliferation. These divergent outcomes highlight that the impact of TFAM loss is context-dependent, shaped by the basal metabolic wiring of each tumor.

U87MG cells display MS and GPM traits, conferring metabolic plasticity that supports survival even when OXPHOS is compromised [34]. In line with this, TFAM silencing induced strong upregulation of *LDHC*, associated with alternative glycolytic pathways and poor prognosis in multiple cancers [35,36,37,38]. LDHC promotes proliferation through cyclin D1 stabilization via PI3K/Akt/GSK3β signaling [36,37], consistent with the enrichment of cell cycle regulators (*MAPK1*/*9*, *HRAS*, *BORA*, *YWHAQ*, *CCNA1*/*2*, *CCNB1*, *AURKA*, and *CDC25A*) in U87MG-siTFAM cells [39,40,41,42,43]. Elevated ROS levels following TFAM knockdown may further fuel proliferation, as oxidative stress can activate cell cycle regulators and drive G1–S progression [44,45,46]. In fact, the oxidative stress caused by high ROS levels may function as a retrograde signaling mechanism, modulating the gene expression of several signaling pathways [47].

Of the TFAM-responsive genes identified, *TRAF2* emerged as a key candidate. TRAF2, a non-canonical NF-κB regulator, promotes survival signaling and activates MAPK cascades [48,49,50,51,52,53]. The gene was upregulated in U87MG-siTFAM cells and inversely correlated with *TFAM* expression. *TRAF2* overexpression has been implicated in proliferation and apoptosis resistance across multiple cancers [54,55,56,57,58], and its association with poor survival in GBM underscores its therapeutic potential. Indeed, *TRAF2* silencing reduces proliferation and increases radiosensitivity in GBM [59] and enhances immunotherapy efficacy in melanoma [60]. The availability of promising TRAF2-targeting agents, such as liquidambaric acid [61] and anti-TNFR2 antibodies currently under clinical evaluation (NCT04752826) [62], strengthens the rationale for TRAF2 as a potential mediator in *TFAM*-low GBM.

Interestingly, TFAM silencing also downregulated androgen receptor (*AR*), consistent with the loss of AR-mediated activation of the *TFAM* promoter. *AR* is typically upregulated in GBM [63] and regulates proliferation via PI3K/Akt-mediated cyclin expression [64,65]. In parallel, AR loss can impair cytoskeletal dynamics by inhibiting PI3K/Akt/RAC1 signaling [66], leading to decreased activity of motility regulators such as *LIMK1* [67,68] and *KIT* [69]. In U87MG-siTFAM cells, multiple AR-linked motility genes (*LIMK1*, *KIT*, *KANK1*, *ANKRD1*, *FZD9*, *SLIT2*, *UNC5C*, and integrins) [70,71,72,73,74,75,76,77] were downregulated, congruent with reduced migration observed in vitro [78]. LOXL2, a key ECM regulator and mediator of cell adhesion [79], was also downregulated, consistent with impaired motility. *LOXL2* loss has been linked to reduced migration in multiple cancer types [80,81,82,83]. Importantly, our group previously associated *LOX* family expression with ECM stiffness and GBM aggressiveness [84], underscoring LOXL2 as another context-dependent vulnerability. An oral LOXL2 inhibitor (GB2064, formerly PAT-1251) is already under clinical evaluation in hematologic malignancy (NCT04679870).

By contrast, TFAM downregulation in A172 cells had limited effects on proliferation and motility. This divergence likely reflects differences in antioxidant capacity between the two cell lines: A172 cells exhibit higher expression of cytosolic superoxide dismutase (*SOD1*), which more effectively buffers ROS, whereas U87MG cells rely predominantly on mitochondrial *SOD2* [85,86]. As a result, A172 cells are better equipped to buffer oxidative stress arising from TFAM loss, thereby mitigating downstream signaling changes and explaining their muted phenotypic response.

Transcriptomic analysis supports this interpretation, revealing that A172-siTFAM cells activate an adaptive molecular program that encompasses mitochondrial membrane regulation, survival signaling, and oxidative stress mitigation. This includes upregulation of genes involved in mitochondrial dynamics and mitotic control (e.g., *E2F1*, *CCND1*), as well as central hubs such as GSK3A and 14-3-3 proteins (coded by *YWHAG*, *YWHAE*, and *YWHAQ*) that bridge cell cycle progression with stress response pathways [40]. Downregulation of OXPHOS components and vesicle trafficking genes suggests a strategic metabolic shift to reduce oxidative burden while maintaining proliferative capacity.

This adaptive resilience contrasts starkly with the vulnerability of U87MG cells, where TFAM loss triggers ROS accumulation, cell cycle acceleration, and motility impairment. Together, these findings underscore that the biological consequences of TFAM dysregulation are highly context-dependent, shaped by the tumor’s intrinsic metabolic wiring and antioxidant defense capacity (Figure 8).

From a translational perspective, the present results suggest that *TFAM*-low, GPM-type GBMs may be particularly susceptible to therapeutic strategies that exploit oxidative stress vulnerabilities or target downstream effectors such as *TRAF2* or *LOXL2*. Conversely, MTC-type GBMs with robust antioxidant defenses may require combination approaches that disrupt their adaptive mitochondrial and stress-response networks.

## 4. Materials and Methods

### 4.1. Tumor Sample Collection and Processing, and TCGA Data

A total of 150 astrocytoma samples encompassing different malignancy grades were collected intraoperatively and immediately snap-frozen in liquid nitrogen by our group at the Division of Neurosurgery, Department of Neurology, School of Medicine, University of São Paulo (FMUSP). This cohort included 26 AG2, 17 AG3, and 84 GBMs. In parallel, an additional 79 tumor specimens—5 AG2, 11 AG3, and 63 GBM—were formalin-fixed and paraffin-embedded (FFPE) for subsequent analyses. Following collection, all frozen tumor tissues were processed for DNA and RNA extraction. To ensure sample integrity, only specimens containing >80% tumor tissue, as determined during quality assessment, were included in downstream analyses. The present study was approved by the Institutional Ethical Committee guidelines at the Hospital das Clinicas of the University of São Paulo School of Medicine (CEP; number 52789221.1.0000.0068). The human samples used in this study were collected in accordance with the same institutional committee and approved for use as a biorepository (830/01), with post-informed consents obtained from all patients included in this study.

### 4.2. TCGA Gene Expression Analysis

Gene expression data from the RNAseq of the glioma dataset of The Cancer Genome Atlas (TCGA) were obtained from the Genomics Commons Data Portal (https://portal.gdc.cancer.gov/ (accessed on 22 July 2022) and normalized using the DESeq package in R. Heatmap visualizations were generated using z-score-normalized RPKM values.

### 4.3. Cell Culture

Glioma cell lines U87MG and A172 (American Type Culture Collection, ATCC; Manassas, VA, USA) were maintained in Dulbecco’s modified Eagle’s medium (DMEM; Life Technologies, Carlsbad, CA, USA), supplemented with 10% fetal bovine serum (FBS; Cultilab, Campinas, Brazil) and antibiotics (Life Technologies). Cells were cultured in a humidified incubator at 37 °C with 5% CO_2_ in air, and routinely tested for mycoplasma contamination.

### 4.4. TFAM Silencing by siRNA

Two small interfering RNA (siRNA) duplex sequences were designed for *TFAM* silencing: siTFAM (5′-rCrUrUrUrArUrUrGrUrGrCrGrArCrGrUrArG-3′/5′-rGrGrArUrCrUrUrCrUrArCrGrUrCrGrCrArC-3′) and a non-targeting control (NTC) siRNA. Sequences were synthesized by Integrated DNA Technologies (IDT, Coralville, IA, USA) and diluted in RNase-free duplex buffer provided by the manufacturer. U87MG and A172 cells (1 × 10^5^ cells/well) were seeded in six-well plates and, after 24 h, transfected with Lipofectamine RNAiMax (Thermo Fisher Scientific, Waltham, MA, USA), together with siTFAM or NTC oligonucleotides at a final concentration of 10 nM for both cell lines. TFAM silencing was evaluated after 2, 4, and 7 days post-transfection by quantitative real-time PCR (qPCR) and Western blot.

### 4.5. TFAM Activation by CRISPRa

To induce *TFAM* overexpression, CRISPR activation (CRISPRa) was performed using two guide oligonucleotides: TF1c1 (Primer 1: 5′CACCGCTAACATCCGGTCAGAT3′, Primer 2: 5′AAACATCTACCGACCGGATGTTAGC3′) and TF2c1 (Primer 1: 5′CACCGAAATCTGCTAACATCCGGT3′, Primer 2: 5′AAACACCGGATGTTAGCAGATTTC3′). Guides were cloned into the LentiSAMPHv2 vector (Addgene plasmid #167934) using the NEBridge^®^ Golden Gate Assembly Kit (New England Biolabs, Ipswich, MA, USA). Lentiviral particles were produced using TF1c1, TF2c1, or empty vector (Vɸ) and transduced into cells with 6 µg/mL of polybrene. Selection with 12 µg/mL blasticidin for approximately 10 days. Two multiplicities of infection (MOI 1 and MOI 2) were tested, and the most efficient was selected. Post-selection, cells were maintained in DMEM supplemented with 10% FBS and 1% penicillin/streptomycin.

### 4.6. Nucleic Acid Extraction and cDNA Synthesis

DNA and RNA were extracted using the AllPrep DNA/RNA Mini Kit (Qiagen, Hilden, Germany) following the manufacturer’s instructions. Concentration and purity were assessed with a NanoDrop ND-1000 spectrophotometer (Thermo Fisher Scientific), with a 260/280 nm ratio of between 1.8 and 2.0 considered suitable for further analysis. For cDNA synthesis, 1 μg of total RNA was reverse-transcribed using random primers, oligo(dT) oligonucleotides, RNasin^®^ ribonuclease inhibitor, and GoScript™ reverse transcriptase (Promega, Madison, WI, USA), then diluted in Tris/EDTA (TE) buffer.

### 4.7. Quantitative PCR and mtDNA Copy Number

*TFAM* expression levels were evaluated by qPCR on a QuantStudio™ 3 system (Thermo Fisher Scientific) using the following primers: TFAM (forward: 5′-CTCCCCCTTCAGTTTTGTGT-3′; reverse: 5′-GCATCTGGGTTCTGAGCTTT-3′) and Hypoxanthine phosphoribosyltransferase (HPRT; forward: 5′-TGAGGATTTGGAAAGGGTGT-3′; reverse: 5′-GAGCACACAGAGGGCTACAA-3′) as the reference gene. Mitochondrial DNA (mtDNA) copy number was assessed using primers for the D-loop region (forward: 5′-TGATGGCTAGGGTGACTTCAT-3′; reverse: 5′-CCTAGCCGTTTACTCAATCCT-3′) and the single-copy nuclear gene hemoglobin beta (HBB) (forward: 5′GTGAAGGCTCATGGCAAGA3′; reverse: 5′-AGCTCACTCAGGTGTGGCAAAG-3′). Reactions were performed utilizing 1 ng/μL of DNA diluted in TE buffer. Reactions were carried out using the Syber Green method, in duplicate, and in two independent experiments. Experiments were repeated whenever the standard deviation of replicates exceeded 0.4. The thermal cycling protocol consisted of 2 min at 50 °C, 10 min at 95 °C, followed by 40 cycles of 15 s at 95 °C, and 1 min at 60 °C. Relative expression levels were calculated using the 2^−ΔΔCt^ method, where: ΔΔCt = (mean Cq_TFAM_ − mean Cq_REFERENCE_) siRNA − (mean Cq_TFAM_ − mean Cq_REFERENCE_) NTC. The calculation was applied to each time point analyzed.

### 4.8. Western Blot

Protein extraction from U87MG and A172 cells, followed by TFAM silencing or activation, was performed using the RIPA lysis buffer (50 mM Tris-HCl, 1% NP-40, 0.25% Na-deoxycholate, 150 mM NaCl, 1 mM EDTA) with protease inhibitor cocktail (Sigma–Aldrich, St. Louis, MO, USA). Concentration was determined by NanoDrop spectrophotometry (A280 measurement). Cell lysates (20 μg of proteins) were resolved on a 4–12% gradient polyacrylamide gel electrophoresis (Thermo Fisher Scientific) using NuPAGE buffer (3-(N-morpholino)propanesulfonic acid–sodium dodecyl sulfate) and electrotransferred to PVDF membranes. Membranes were blocked for 1 h in 5% of non-fat dry milk (NFDM) prepared in Tris-buffered saline (TBS), then incubated overnight at 4 °C with primary antibodies: anti-TFAM (ab47517, 1:500; Abcam, Cambridge, MA, USA) and anti-β-actin (A2228, 1:50,000, Sigma-Aldrich) as a loading control. After washing, membranes were incubated for 2 h with horseradish peroxidase-conjugated anti-rabbit and anti-mouse IgG secondary antibodies (Sigma-Aldrich, 1:1000). Protein bands were visualized using the Clarity Western ECL Blotting Substrate (BioRad Laboratories, Hercules, CA, USA) and imaged on ImageQuant LAS 4000 system (GE Healthcare, Pittsburgh, PA, USA). Densitometric analysis of immunoreactive bands was performed using ImageJ/Fiji software (version 1.54f, National Institutes of Health, Bethesda, MD, USA) obtained from imagej.nih.gov/ij/download/ (accessed on 22 October 2025) and normalized to NTC values for each time point.

### 4.9. Immunohistochemistry

TFAM immunohistochemistry was performed using the Novolink polymer detection system (Novocastra, Newcastle-upon-Tyne, UK) according to the manufacturer’s instructions. Antigen retrieval was carried out in citrate buffer (pH 6.0) for 3 min at 122 °C using an electric pressure cooker (BioCare Medical, Walnut Creek, CA, USA). Tissue sections were incubated overnight at 4 °C with primary antibody anti-TFAM (ab47517, 1:600; Abcam), followed by visualization with diaminobenzidine (DAB) and counterstaining with Harris hematoxylin for nuclear visualization. Digital photomicrographs were captured with 400× magnification across five distinct fields. Background illumination was normalized using NIS-Elements Viewer software version 6.10.01 (Nikon Instruments, Tokyo, Japan) to ensure a neutral color background. Images were subsequently color-deconvoluted using ImageJ/Fiji software version 1.54g. The intensity label score was calculated by measuring the difference between the total absence of staining (white background) and the average DAB intensity staining.

### 4.10. Cell Viability and Proliferation

Cell viability was assessed in U87MG and A172 cells following TFAM silencing (siTFAM) or control treatment (NTC). Cells (1 × 10^3^/well) were seeded in 96-well plates. At days 2, 4, and 7 post-transfection, cells were incubated with PrestoBlue reagent (Thermo Fischer Scientific) for 2 h at 37 °C in a humidified atmosphere containing 5% CO_2_. Fluorescence intensity was measured using GloMax^®^ 96 Microplate Luminometer (Promega) with excitation at 570 nm and emission at 610 nm. Background fluorescence from DMEM supplemented with 10% FBS was measured for each plate and subtracted from the sample readings.

For proliferation, U87MG cells (2.5 × 10^4^/well) transduced with TF1c1, TF2c1, and Vɸ lentivirus, were seeded in 24-well plates. Cells were trypsinized, stained with Tripan blue, and counted every 24 h for up to 72 h using a Countess 3 automated cell counter (Thermo Fisher Scientific). The total number of live and dead cells was quantified and analyzed using GraphPad Prism^®^ software, version 8.

### 4.11. Cell Cycle Analysis

Cell cycle phase distribution in U87MG and A172 cell lines treated with siTFAM or NTC was analyzed by flow cytometry. At day 3 post-transfection, cells were synchronized in FBS-free DMEM supplemented with 0.5% bovine serum albumin for 24 h. Following synchronization, the cell medium was replaced with standard culture medium (DMEM supplemented with 10% FBS) for an additional 24 h. Cells were fixed with cold ethanol in increasing concentrations (25%, 50%, 75%, and 90%), washed, and stained with propidium iodide (PI; Thermo Fisher Scientific). Fluorescence was measured using a FACSCanto flow cytometer (BD Biosciences, East Rutherford, NJ, USA), and cell cycle profiles were analyzed using FlowJo software version 10 (FlowJo, LLC, Ashton, OR, USA) via the cell cycle interface.

### 4.12. Cell Motility Analyses

The cell motility ability of U87MG and A172-siTFAM cells was evaluated through wound healing and transwell migration. Wound healing assay was performed by seeding a total of 8 × 10^4^ cells in 48-well plates to reach full confluence. After 24 h, the culture medium was removed, and a scratch was made in the monolayer using a 10 μL micropipette tip to create a cell-free area (“wound”). Wells were washed with phosphate-buffered saline (PBS) to remove detached cells, and fresh medium containing 1% FBS was added to minimize cell proliferation. Scratch closure was monitored at 0, 12, and 24 h using reference points marked on the plate bottom of each well to ensure consistent field selection. Two independent experiments were performed in triplicate. The gap area (μm^2^) was measured using ImageJ/Fiji software, and the percentage of invaded area was calculated by subtracting the remaining gap area from the initial wound area (100%).

Transwell migration assay was performed using Boyden chamber inserts with 8 μm pore size (Corning Incorporated, Corning, NY, USA). A total of 2.5 × 10^4^ cells were suspended in 0.5 mL of DMEM supplemented with 1% of FBS and seeded into the upper chamber. Inserts were placed in wells with 500 μL of DMEM and 10% of FBS as a chemoattractant and incubated overnight at 37 °C in a humidified atmosphere with 5% CO_2_. Non-migrating cells were removed from the upper surface of the membrane. Migrated cells were fixed with 4% paraformaldehyde for 20 min at 37 °C, washed with water, stained with 0.2% crystal violet in 20% methanol, and imaged using inverted photomicroscopy at 10× magnification. Quantification was performed by counting all stained cells across the membrane from two independent experiments.

### 4.13. Cell Adhesion Assay

Cells were incubated for 2 h in DMEM supplemented with 1% FBS at 37 °C in 5% CO_2_. After trypsinization, 5 × 10^4^ cells were seeded into 96-well plates and allowed to adhere for 3 h under the same conditions. Non-adherent cells were removed by gently washing with PBS (three times), and the viability of attached cells was quantified using PrestoBlue Cell Viability Reagent (Thermo Fisher Scientific) after 2 h incubation. Fluorescent measurements were made according to the protocol in the “Cell Viability Assay” section. All data were plotted and analyzed using GraphPad Prism^®^ version 8.

### 4.14. General ROS Detection

Reactive oxygen species (ROS) levels were assessed in U87MG and A172 cells treated with siTFAM or NTC using the cell-permeant probe 2′,7′-dichlorodihydrofluorescein diacetate (H_2_DCFDA; Thermo Fisher Scientific). Upon intracellular ROS generation, the non-fluorescent H_2_DCFDA was oxidized to the highly fluorescent compound 2′,7′-dichlorofluorescein (DCF), which was detected by flow cytometry using a FACSCanto (BD Biosciences). Data were analyzed by FlowJo software version 10. Non-stained cells were used as negative controls, while parental cells treated with 1 μM hydrogen peroxide for 1 h at 37 °C in 5% CO_2_ were used as positive controls.

### 4.15. Mitochondrial Superoxide Detection

Mitochondrial superoxide production was assessed on day 4 post-transfection in U87MG and A172 cells from both siTFAM and NTC groups. The analysis was performed in triplicate for each condition across two independent experiments. Cells were stained using the MitoSOX^TM^ Red Mitochondrial Superoxide Indicator kit (Thermo Fisher Scientific) following the manufacturer’s instructions. Fluorescence was measured by flow cytometry using FACSCanto (BD Biosciences), and data were analyzed by FlowJo software version 10. Non-stained cells were used as negative controls, and parental cells treated with 1 μM hydrogen peroxide for 1 h at 37 °C in 5% CO_2_ were used as positive controls.

### 4.16. RNA Sequencing and Bioinformatics

RNA-Seq was conducted to identify differentially expressed genes upon TFAM silencing in U87MG and A172 cells. Total RNA of each sample was used in duplicate, and libraries were constructed using the Illumina Stranded Total RNA Prep with Ribo-Zero Plus kit (Illumina, San Diego, CA, USA), as per the manufacturer’s instructions. Library quality was assessed using the TapeStation 2200 system with D1000 ScreenTape (Agilent Technologies, Santa Clara, CA, USA), and quantification was performed by qPCR using the Kapa Library Quantification Kit (Kapa Biosystems, Roche, Pleasanton, CA, USA). DNA libraries were pooled and sequenced on an Illumina HiSeq 2500 platform (Illumina) at the SELA Facility Core, School of Medicine, University of São Paulo, generating an average of 51 million reads per sample. Quality control of raw reads was performed using the FASTQC software version 0.12.0. Reads were aligned to the GRCh38 human genome using the STAR aligner, and gene-level read counts were obtained using featureCounts. Normalization of count data was performed using the trimmed mean M-values (TMM) method implemented in the edgeR package version 3.21. Gene expression levels were calculated using reads per kilobase per million (RPKM) and log counts per million (logCPM). Differential gene expression between the siTFAM and NTC groups was determined using the limma package from R-Bioconductor. Gene-set enrichment and pathway analysis were performed using WebGestalt 2024 (Web-Based Gene Set Analysis Toolkit), applying over-representation analysis (ORA) with the Gene Ontology (GO) Biological Process database. Enriched pathways were further refined using REVIGO to eliminate redundant terms, retaining only those with a dispensability score ≤0.05.

### 4.17. Statistical Analysis

Statistical analysis and graphical representations were performed on SPSS software version 23.0 (IBM, Armonk, NY, USA) and GraphPad Prism^®^ 8.0 version (GraphPad Software Inc., San Diego, CA, USA) software. Normality of the data distribution was assessed using the Kolmogorov–Smirnov test. For non-parametric data, group differences were evaluated using the Kruskal–Wallis test followed by Dunn’s post hoc test. Pairwise comparisons between two groups with non-parametric distribution were evaluated using the Mann–Whitney test. The normality test was performed for all functional experiments, and since the data distribution was found to be parametric, the subsequent tests used were also parametric. For cell viability and cell cycle experiments, the two-way analysis of variance test (ANOVA) was applied to compare multiple groups, followed by Tukey’s post hoc test. Two-way ANOVA with Bonferroni correction was used for multiple gene expression analysis, cell motility assays, general and mitochondrial superoxide production comparisons between siTFAM and NTC groups, and between Vϕ and TF1c1 or TF2c1. Correlations between gene expression values were determined using Spearman’s rho test. Variable discrimination was evaluated using receiver operator characteristics (ROC) curve analysis, including area under the curve (AUC) and asymptotic significance. The Kaplan–Meier estimator was employed to assess the impact on overall survival (OS) in GBM-GPM and MTC cases, focusing on those genes with significant differential expression between “*TFAM* or *AR* high” and “*TFAM* or *AR* low” groups. Sample stratification into high and low expression categories was based on ROC curve thresholds, and survival differences were analyzed by the log-rank test. Statistical significance was defined as *p* ≤ 0.05.

## 5. Conclusions

The current findings identify TFAM as a context-dependent regulator of glioma cell phenotype, with its loss driving proliferative, ROS-prone states in GPM GBM cells but eliciting adaptive stress-resilient programs in MTC-type cells. By integrating phenotypic assays with transcriptomic profiling, distinct downstream effectors were revealed—including TRAF2 and LOXL2—that connect TFAM dysregulation to cell cycle control, metabolic reprogramming, and tumor invasiveness. These findings not only position TFAM as a potential biomarker for glioma stratification but also highlight therapeutic opportunities tailored to metabolic subtype, such as targeting oxidative stress vulnerabilities in GPM-type GBMs or disrupting adaptive mitochondrial networks in MTC-type tumors. Future work should explore pharmacological modulation of TFAM and its effectors as a precision medicine strategy in high-grade gliomas.

## Figures and Tables

**Figure 1 ijms-26-10446-f001:**
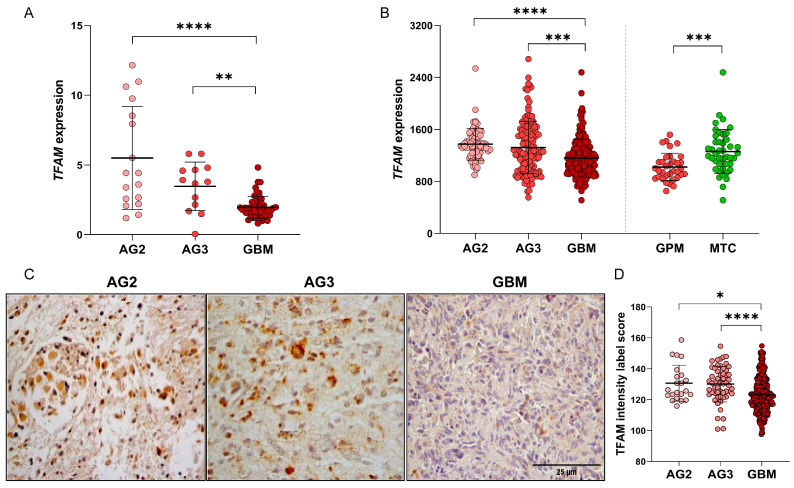
Analysis of TFAM expression in astrocytomas: (**A**) q-PCR analysis of *TFAM* expression in astrocytomas of different grades of malignancy showing a significant decrease in *TFAM* expression with increasing malignancy. (**B**) In silico analysis of the TCGA astrocytoma database confirming a significant decrease in *TFAM* expression in GBM relative to AG2 and AG3. GBM subtype according to the metabolic axis, based on Garofano pathway-based classification, shows lower *TFAM* expression in GPM than in MTC. (**C**) Representative images of TFAM immunostaining in AG2, AG3, and GBM showing lower expression in GBM (magnification 400×). (**D**) Immunolabelling analysis showing TFAM intensity label score and a significant decrease in GBM cases compared to AG2 and AG3 on Dunn’s post hoc test. Dots represent each case analyzed, and the line represents mean ± SD. AG2, astrocytoma grade 2; AG3, astrocytoma grade 3; GBM, glioblastoma; MTC, mitochondrial subtype; GPM, glycolytic/plurimetabolic subtype; SD, standard deviation; TCGA, The Cancer Genome Atlas. Statistically significant differences represented by asterisks: * *p* < 0.05, ** *p* < 0.01, *** *p* < 0.001, **** *p* < 0.0001.

**Figure 2 ijms-26-10446-f002:**
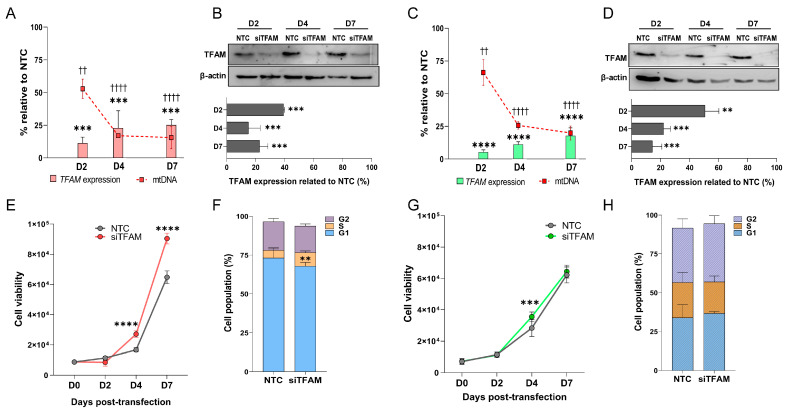
Transient TFAM silencing by siRNA: (**A**,**C**) RT-qPCR analysis of *TFAM* mRNA expression (bars) and mtDNA copy number (squares connected by lines) in U87MG (**A**) and A172 (**C**) cells at 2 (D2), 4 (D4), and 7 (D7) days post-transfection, relative to non-targeting control (NTC). *HPRT* was used as the reference gene. The experiment was performed using biological duplicates. (**B**,**D**) Representative Western blots of TFAM protein in U87MG (**B**) and A172 (**D**) cells under NTC conditions and at 2, 4, and 7 days following TFAM silencing. Asterisks represent the differences observed in *TFAM* expression, and crosses represent differences observed in mtDNA copy number. β-actin was used as the loading control (TFAM: 29 kDa; β-actin: 42 kDa). Densitometric quantification was performed using ImageJ, with TFAM bands normalized to β-actin and expressed relative to NTC. Data are presented as mean ± SD from two independent experiments. (**E**,**G**) Cell proliferation curves for U87MG (**E**) and A172 (**G**) cells following TFAM silencing. Cell viability was normalized to day 0 (D0) and measured at 2, 4, and 7 (D2, D4, and D7) days post-transfection in the siTFAM and NTC groups. Data points represent mean ± SD of replicates. Cell viability was performed using technical octuplicates and biological duplicates. (**F**,**H**) Cell cycle distribution assessed by flow cytometry after propidium iodide staining at day 4 post-transfection. In U87MG cells (**F**), siTFAM treatment induced a shift from G0/G1 to S phase, whereas A172 cells (**H**) showed no significant changes. Bars indicate the percentage of cells in G0/G1 (bottom), S (middle), and G2/M (top) phases. The experiment was conducted using technical triplicate and biological duplicate. SD, standard deviation. Statistically significant differences represented by asterisks: ** *p* < 0.01, *** *p* < 0.001, **** *p* < 0.0001; †† *p* < 0.01, †††† *p* < 0.0001.

**Figure 3 ijms-26-10446-f003:**
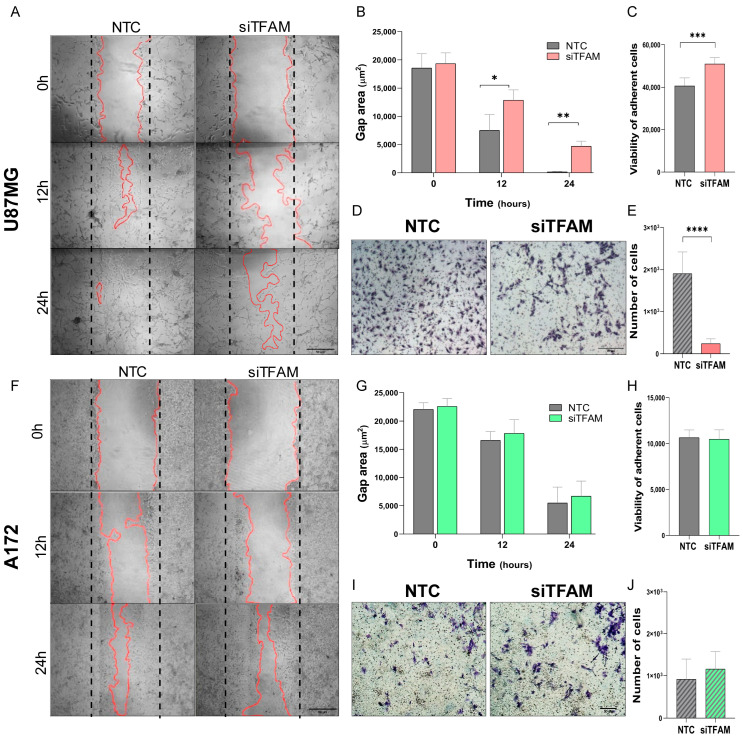
Impact of TFAM silencing on cell motility: (**A**,**F**) Representative photomicrographs of wound healing assay in U87MG (**A**) and A172 (**F**) cells, comparing non-targeting control (NTC) and TFAM-silenced (siTFAM) groups at 0, 12, and 24 h after scratch (performed at day 4 post-transfection). Red lines indicate the wound edge. Images were captured at 10× magnification. (**B**,**G**) Quantification of wound gap area (μm^2^) using ImageJ. Data are expressed as mean ± SD of replicates. U87MG-siTFAM cells (**B**) exhibited significantly reduced migration, as evidenced by larger wound areas at 12 and 24 h, whereas no differences were observed in A172-siTFAM cells (**G**) compared to NTC. (**C**,**H**) Quantification of adherent cells in the short-term adhesion assay (3 h). TFAM silencing increased adhesion in U87MG (**C**) but had no effect in A172 (**H**) cells compared with NTC. Bars represent mean ± SD of replicates. (**D**,**I**) Representative photomicrographs of Boyden’s chamber migration assay for U87MG (**D**) and A172 (**I**) cells (NTC vs. siTFAM), acquired at 10× magnification. (**E**,**J**): Quantification of migratory cells in Boyden’s chamber assay for U87MG (**E**) and A172 (**J**). Data represent mean ± SD of replicates. A significant reduction in migration was observed in U87MG-siTFAM cells, while no differences were detected in A172-siTFAM cells. Cell motility assays were performed in technical triplicate and biological duplicate. SD, standard deviation. Statistically significant differences represented by asterisks: * *p* < 0.05, ** *p* < 0.01, *** *p* < 0.001, **** *p* < 0.0001.

**Figure 4 ijms-26-10446-f004:**
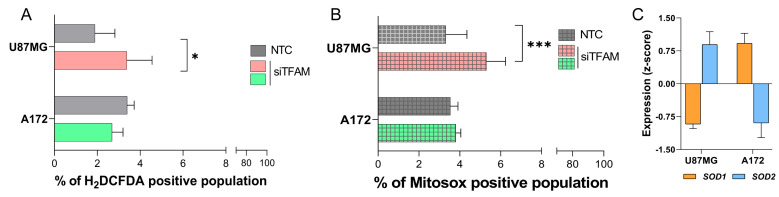
Cellular oxidative stress following TFAM silencing: (**A**) Intracellular ROS levels measured by H_2_DCFDA conversion to green-fluorescent product and detected in FITC channel by flow cytometry. Histograms show the percentage of FITC-positive cells (mean ± SD) in U87MG and A172 cells transfected with siTFAM compared to NTC. (**B**) Mitochondrial superoxide production assessed using MitoSOX and quantified as a percentage of PE-positive cells. Bars represent mean ± SD for siTFAM versus NTC in U87MG and A172 cells. ROS measurements were evaluated in technical triplicate and biological duplicate. (**C**) *SOD1* and *SOD2* expression levels in U87MG and A172 cells normalized by z-score. SD, standard deviation. Statistically significant differences represented by asterisks: * *p* < 0.05, *** *p* < 0.001.

**Figure 5 ijms-26-10446-f005:**
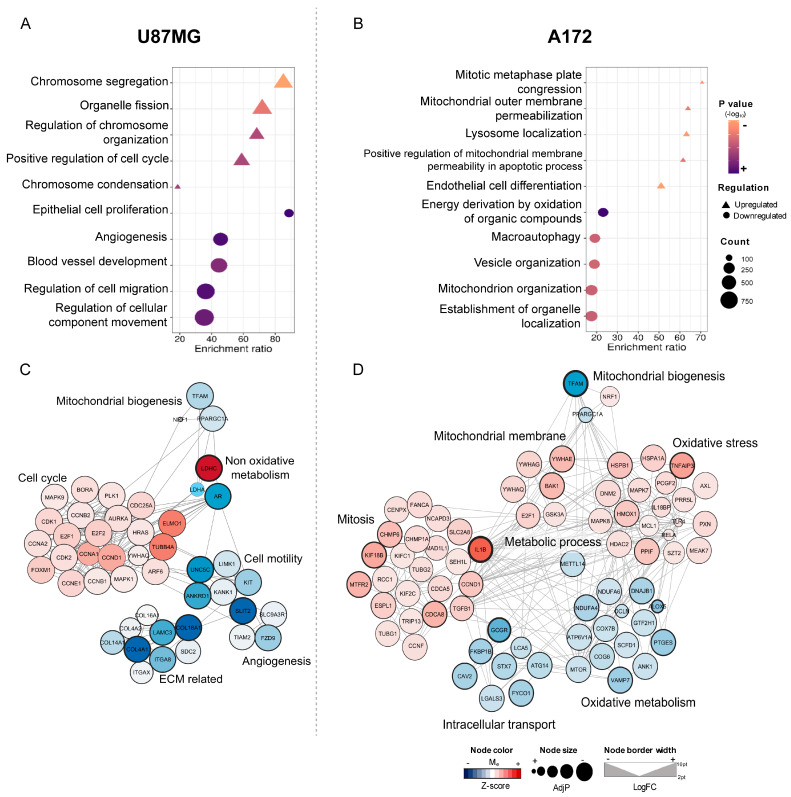
Enrichment analysis of RNA-seq from TFAM-silenced U87MG and A172 cell lines: (**A**,**B**) Dot plots showing top five Gene Ontology (GO) Biological Process terms among differentially expressed genes in U87MG-siTFAM (**A**) and A172-siTFAM (**B**) cells. Shapes indicate direction of regulation: triangles = upregulated processes; circles = downregulated processes. The color gradient reflects statistical significance (-log_10_ *p*-value; darker = more significant). Symbol size corresponds to the number of genes in each term. (**C**,**D**) Gene-pathway interaction networks for U87MG-siTFAM (**C**) and A172-siTFAM (**D**), illustrating connections between enriched GO terms and their associated genes. Node color represents z-score of expression; node size reflects adjusted *p*-value significance (larger = more significant); border width indicates magnitude of log_2_ fold change. These networks highlight the interplay between upregulated and downregulated pathways, and identify central regulators driving transcriptional responses to TFAM silencing in each cell line.

**Figure 6 ijms-26-10446-f006:**
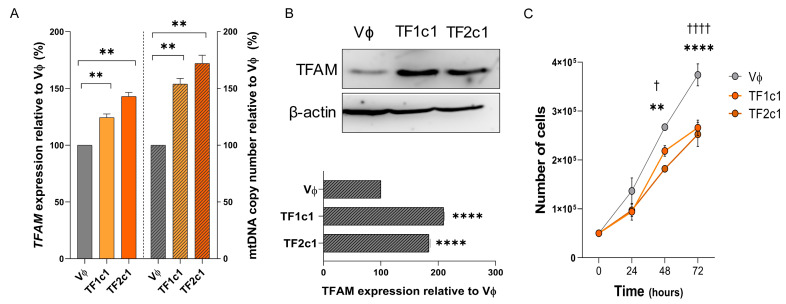
TFAM overexpression in the U87MG cell line: (**A**) *TFAM* expression levels (left panel) and mitochondrial DNA (mtDNA) copy number (right panel) in U87MG cells overexpressing *TFAM* (TF1c1, TF2c1) compared with empty vector (Vϕ). Data are presented as mean ± SD, expressed as percentages relative to Vϕ. (**B**) Representative Western blot analysis for TFAM protein expression in U87MG cells. β-actin (42 kDa) was used as a loading control; the TFAM protein band appears at 29 kDa. Densitometric quantification shows mean ± SD of TFAM protein levels relative to Vϕ, normalized to β-actin using ImageJ. (**C**) Cell proliferation assay comparing control and TFAM-overexpressing U87MG cells. Bars represent the number of viable cells (mean ± SD). Asterisks represent the differences observed between Vϕ x TF1c1, and crosses represent differences between Vϕ x TF2c1. SD, standard deviation. Statistically significant differences represented by asterisks: ** *p* < 0.01, **** *p* < 0.0001; † *p* < 0.05, †††† *p* < 0.0001.

**Figure 7 ijms-26-10446-f007:**
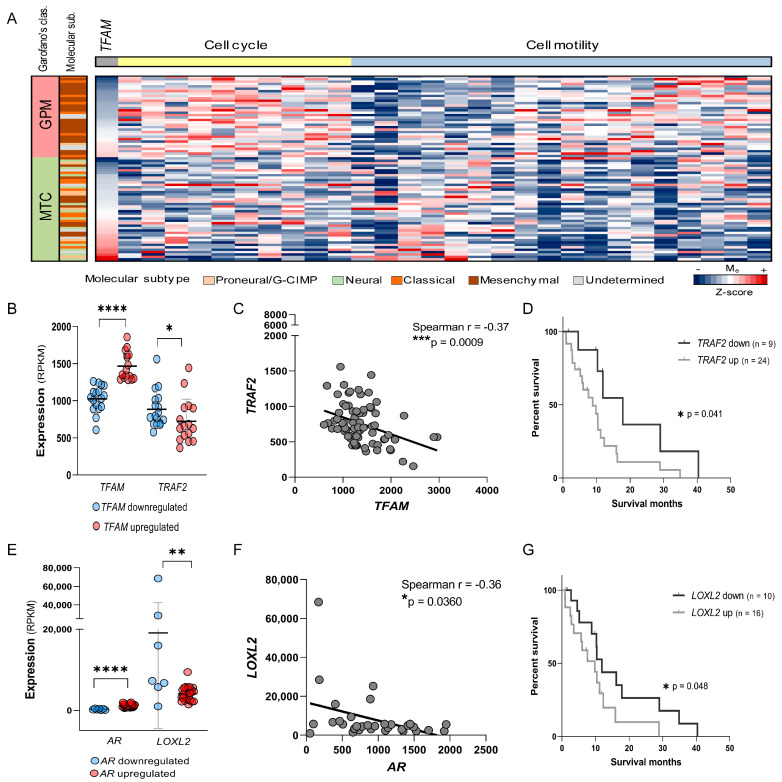
TFAM expression correlates with cell cycle and motility genes in GBM: (**A**) Heatmap of genes associated with cell cycle and motility in GBM glycolytic/plurimetabolic (GPM, pink) and mitochondrial (MTC, green) subtypes, ordered by *TFAM* expression (z-score). (**B**,**E**) Scatter plots of differential gene expression between *TFAM*-downregulated (blue) and *TFAM*-upregulated (red) groups (**B**), and between *AR*-downregulated (blue) and *AR*-upregulated (red) groups (**E**). Each dot represents an individual case; lines show mean ± SD. (**C**,**F**) Spearman correlation between *TFAM* and *TRAF2* expression (**C**) and between *AR* and *LOXL2* expression (**F**) in GBM-GPM and MTC cases. (**D**,**G**) Kaplan–Meier survival analysis of *TRAF2* (**D**) and *LOXL2* (**G**) expression in GBM-GPM and MTC subtypes. Seven cases were censored in both analyses. SD, standard deviation. Statistically significant differences represented by asterisks: * *p* < 0.05, ** *p* < 0.01, *** *p* < 0.001, **** *p* < 0.0001.

**Figure 8 ijms-26-10446-f008:**
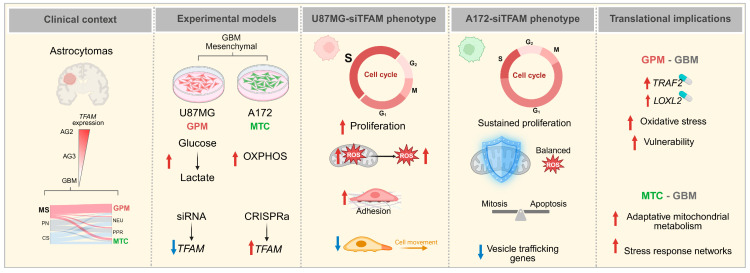
Schematic representation of TFAM’s role in GBM experimental models and its impact on GBM metabolic subtypes. TFAM expression decreases with astrocytoma grade, reaching its lowest levels in GPM-GBM, a subtype associated with poor prognosis. Functional and transcriptomic analyses in mesenchymal GBM cell lines revealed that TFAM silencing in GPM-type U87MG cells promotes proliferation, S-phase entry, ROS accumulation, and adhesion, while impairing motility. By contrast, MTC-type A172 cells activated an adaptive program involving antioxidant defenses with balanced ROS, and survival signaling, buffering against oxidative stress and limiting phenotypic change. These findings highlight TFAM as a context-dependent regulator of glioma behavior and a potential therapeutic node. In TCGA-GPM cases, high *TRAF2* expression—a cell cycle regulator—was correlated with *TFAM* and associated with poor prognosis. Similarly, *LOXL2* expression—a cell motility regulator—was correlated with *AR* expression and directly impacted by TFAM silencing, also being associated with poor prognosis. Together, these findings may promote combinatory therapies in GPM-GBM. AG2, astrocytoma Grade 2; AG3, astrocytoma Grade 3; GBM, glioblastoma; MS, mesenchymal; PN, proneural; CS, classical; GPM, glycolytic/plurimetabolic; NEU, neuronal; PPR, proliferative/progenitor; MTC, mitochondrial; OXPHOS, oxidative phosphorylation; ROS, reactive oxygen species; TCGA, The Cancer Genome Atlas; AR, androgen receptor. Created with Biorender.com.

## Data Availability

The sequencing data are available upon request from the corresponding author, and the GEO study ID will be reported as soon as the data are deposited in the NCBI-GEO platform.

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
