# Peer review of "TFAM Loss Induces Oxidative Stress and Divergent Phenotypes in Glioblastoma Metabolic Subtypes"

_ijms, 2025, doi:10.3390/ijms262110446_

Round 1

Reviewer 1 Report

Comments and Suggestions for Authors

This manuscript investigates the role of mitochondrial transcription factor A (TFAM) in glioblastoma (GBM), highlighting its subtype-specific roles in proliferation, redox balance, and motility. The authors integrate transcriptomic, functional, and patient-derived data to reveal divergent consequences of TFAM depletion in glycolytic / plurimetabolic (GPM) and mitochondrial (MTC) metabolic subtypes.  The topic is increasingly interesting in metabolic stratification of GBM and mitochondrial biology. The study provides novel and compelling data, particularly regarding the redox-sensitive phenotype in GPM cells and compensatory survival responses in MTC cells.

However, a few areas require clarification and improvement, including stronger justification for the cell line models used, mechanistic insight into TFAM’s role, and additional discussion of in vivo or clinical relevance.

Specifically,

  1. Justification of Cell Line Models: U87MG and A172 are widely used but do not fully recapitulate primary GBM metabolism. The classification into GPM and MTC subtypes must be explicitly justified. Please reference or summarize prior metabolic profiling (as per ref. [19]) used to categorize these lines.
  2. Mechanistic Insight into TFAM Function (Figures 5): The link between TFAM depletion and nuclear gene expression changes (e.g., AR, LDHC, TRAF2) remains correlative. In Figures 5A and 5B, enrichment in cell cycle, mitochondrial, and motility pathways is shown, but causality is not established. Please discuss whether these transcriptional effects arise via retrograde signaling due to mitochondrial dysfunction (e.g., ROS, mtDNA loss), or whether TFAM may have nuclear functions. Consider referencing literature on mitochondrial-nuclear communication in GBM.
  3. Statistical Methods and Multiple Testing (Figures 5, 7): Please clarify how multiple comparisons were corrected for GO and pathway enrichment.. For correlation analyses in Figure 7C (Spearman), clarify the number of samples used and correction method.
  4. Gene/Protein Nomenclature: Italicize all gene symbols (TFAM, LDHC, TRAF2, AR, LOXL2, etc.) and use regular font for protein names
  5. Figure Clarity and Accessibility: In Figures 2 and 3, clarify axis labels and significance markers. Ensure each figure panel is labeled in the text clearly, and legends include exact n values and replicate types.

Author Response

We sincerely thank the reviewers and the editor for their constructive feedback, which has improved the clarity and rigor of the manuscript. 
Below we provide point-by-point responses.

Reviewer #1

Comments and Suggestions for Authors:

This manuscript investigates the role of mitochondrial transcription factor A (TFAM) in glioblastoma (GBM), highlighting its subtype-specific roles in proliferation, redox balance, and motility. The authors integrate transcriptomic, functional, and patient-derived data to reveal divergent consequences of TFAM depletion in glycolytic / plurimetabolic (GPM) and mitochondrial (MTC) metabolic subtypes.  The topic is increasingly interesting in metabolic stratification of GBM and mitochondrial biology. The study provides novel and compelling data, particularly regarding the redox-sensitive phenotype in GPM cells and compensatory survival responses in MTC cells.

However, a few areas require clarification and improvement, including stronger justification for the cell line models used, mechanistic insight into TFAM’s role, and additional discussion of in vivo or clinical relevance.

Specifically,

  1. Justification of Cell Line Models: U87MG and A172 are widely used but do not fully recapitulate primary GBM metabolism. The classification into GPM and MTC subtypes must be explicitly justified. Please reference or summarize prior metabolic profiling (as per ref. [19]) used to categorize these lines.

We appreciate the reviewer’s comment regarding the rationale for using U87MG and A172 cell lines and the justification of their classification into GPM and MTC metabolic subtypes.

In our previous study [1], we systematically characterized the metabolic profiles of these two GBM cell lines using Gene Set Variation Analysis (GSVA) [2] to compute enrichment scores across the full set of metabolic marker genes defined in Garofano’s GBM metabolic classification. This analysis revealed that U87MG exhibits a “glycolysis-prone and metabolically plastic” phenotype (GPM subtype), with the capacity to activate multiple metabolic pathways, including those independent of mitochondrial oxidative metabolism. In contrast, A172 cells display a predominantly mitochondrial metabolism–dependent phenotype (MTC subtype), characterized by higher reliance on oxidative phosphorylation.

This established metabolic distinction provided the rationale for selecting these two well-characterized and widely used GBM models, as they represent the two major metabolic subtypes described by Garofano et al. [3]. We have now clarified this rationale and summarized the key differences in mitochondrial-based pathways between GPM and MTC subtypes in the revised text (lines 81–83).

[1] Moretti, I.F.; Lerario, A.M.; Sola, P.R.; Macedo-da-Silva, J.; Baptista, M.D.S.; Palmisano, G.; Oba-Shinjo, S.M.; Marie, S.K.N. GBM Cells Exhibit Susceptibility to Metformin Treatment According to TLR4 Pathway Activation and Metabolic and Antioxidant Status. Cancers (Basel) 2023, 15, 587.

[2] Hänzelmann, S.; Castelo, R.; Guinney, J. GSVA: Gene set variation analysis for microarray and RNA-seq data. BMC Bioinformatics. 2013, 14, 7.

[3] Garofano, L.; Migliozzi, S.; Oh, Y.T.; D’Angelo, F.; Najac, R.D.; Ko, A.; Frangaj, B.; Caruso, F.P.; Yu, K.; Yuan, J.; et al. Pathway-based classification of glioblastoma uncovers a mitochondrial subtype with therapeutic vulnerabilities. Nat. Cancer 2021, 2, 141-156.

  1. Mechanistic Insight into TFAM Function (Figures 5): The link between TFAM depletion and nuclear gene expression changes (e.g., AR, LDHC, TRAF2) remains correlative. In Figures 5A and 5B, enrichment in cell cycle, mitochondrial, and motility pathways is shown, but causality is not established. Please discuss whether these transcriptional effects arise via retrograde signaling due to mitochondrial dysfunction (e.g., ROS, mtDNA loss), or whether TFAM may have nuclear functions. Consider referencing literature on mitochondrial-nuclear communication in GBM.

We appreciate the reviewer’s insightful comment regarding the mechanistic link between TFAM depletion and nuclear gene expression changes. The causal relationship indeed remains an open question, as both direct nuclear roles of TFAM and indirect effects through mitochondrial retrograde signaling may contribute.

Emerging evidence supports the existence of nuclear TFAM functions beyond its canonical mitochondrial role. Nuclear-localized TFAM has been reported to regulate its own transcription by inhibiting NRF-1 activity [1], as well as to modulate the expression of BIRC5 and CDKN1A in prostate cancer cells [2] and Serca2 in rat cardiomyocytes [3]. Moreover, TFAM’s HMG-box domains enable its binding to damaged nuclear DNA [4,5], potentially facilitating DNA repair through interaction with p53 and other repair factors [6]. However, to the best of our knowledge, direct evidence for nuclear TFAM activity in glioblastoma (GBM) remains lacking.

Alternatively, the transcriptional effects we observe upon TFAM depletion could arise from mitochondrial dysfunction–induced retrograde signaling. TFAM loss leads to mtDNA depletion and increased ROS production, which are well-established triggers of mitochondrial-nuclear communication. Elevated ROS can activate key signaling pathways such as PI3K/AKT, NF-κB, and stress-responsive transcriptional programs that regulate metabolism, cell cycle, and motility [7,8]. Such retrograde signaling cascades are known to influence gene expression networks relevant to GBM pathogenesis, including those governing cellular proliferation and invasion.

To reflect this, we have revised the manuscript to explicitly discuss ROS-mediated retrograde signaling as a potential mechanism linking TFAM depletion to altered nuclear gene expression (lines 407–409).

[1] Lee, E.J.; Kang, Y.C.; Park, W.H.; Jeong, J.H.; Pak, Y.K. Negative Transcriptional Regulation of Mitochondrial Transcription Factor A (TFAM) by Nuclear TFAM. Biochem. Biophys. Commun. 2014, 450, 166-171.

[2] Han, B.; Izumi, H.; Yasuniwa, Y.; Akiyama, M.; Yamaguchi, T.; Fujimoto, N.; Matsumoto, T.; Wu, B.; Tanimoto, A.; Sasaguri, Y.; et al. Human Mitochondrial Transcription Factor A Functions in Both Nuclei and Mitochondria and Regulates Cancer Cell Growth. Biochem. Biophys. Res. Commun. 2011, 408, 45-51.

[3] Watanabe, A.; Arai, M.; Koitabashi, N.; Niwano, K.; Ohyama, Y.; Yamada, Y.; Kato, N.; Kurabayashi, M. Mitochondrial transcription factors TFAM and TFB2M regulate Serca2 gene transcription. Cardiovasc Res. 2011, 90(1): 57-67.

[4] Kozhukhar, N.; Alexeyev, M.F. 35 Years of TFAM Research: Old Protein, New Puzzles. Biology (Basel). 2023, 12(6): 823.

[5] Chew, K.; Zhao, L. Interactions of Mitochondrial Transcription Factor A with DNA Damage: Mechanistic Insights and Functional Implications. Genes 2021, 12(8): 1246.

[6] Wong, T.S.; Rajagopalan, S.; Freund, S.M.; Rutherford, T.J.; Andreeva, A.; Townsley, F.M.; Petrovich, M.; Fersht, A.R. Biophysical characterizations of human mitochondrial transcription factor A and its binding to tumor suppressor p53. Nucleic Acids Res. 2009, 37(20): 6765-83.

[7] Picard, M.; Shirihai, O.S. Mitochondrial signal transduction. Cell Metab. 2022, 34(11): 1620-1653.

[8] Bellanti, F.; Coda, A.R.D.; Trecca, M.I.; Lo Buglio, A.; Serviddio, G.; Vendamiale, G. Redox Imbalance in Inflammation: The interplay of oxidative and reductive stress. Antioxidants (Basel). 2025, 14(6): 656.

  1. Statistical Methods and Multiple Testing (Figures 5, 7): Please clarify how multiple comparisons were corrected for GO and pathway enrichment. For correlation analyses in Figure 7C (Spearman), clarify the number of samples used and correction method.

We thank the reviewer for this important point regarding statistical analyses and correction for multiple testing.

For the pathway enrichment analysis (Figures 5A and 5B), we used the list of differentially expressed genes identified in siTFAM cells compared with the non-targeting control (NTC). Only genes with an adjusted p-value ≤ 0.05 (Benjamini–Hochberg correction for multiple testing) were included. This gene list was analyzed in WebGestalt using the Gene Ontology (GO) Biological Process database. The resulting enriched GO terms were subsequently processed in REVIGO to remove redundant or overlapping categories (e.g., “cell cycle” and “cell division”), ensuring a more concise and interpretable representation of biological pathways.

Because the most pronounced transcriptional alterations were observed in GPM cases, all analyses were performed specifically on the MS-GBM subtype (n = 33), which includes the highest proportion of GPM tumors (45% compared with 22% among MTC cases). For visualization clarity, the heatmap includes all molecular subtypes to highlight the clustering pattern across the cohort. The exact number of cases used and the analytical details have been clarified in the revised text (lines 334–337).

For the correlation analyses presented in Figure 7C, Spearman correlation coefficients were calculated based on data from the same 33 MS-GBM samples. All correlation p-values were adjusted for multiple comparisons using the Benjamini–Hochberg false discovery rate (FDR) method.

  1. Gene/Protein Nomenclature: Italicize all gene symbols (TFAM, LDHC, TRAF2, AR, LOXL2, etc.) and use regular font for protein names

We thank the reviewer for this helpful observation. We have carefully reviewed the entire manuscript and ensured that all gene symbols (e.g., TFAM, LDHC, TRAF2, AR, LOXL2, etc.) are italicized, while protein names are presented in regular font, in accordance with standard nomenclature guidelines.

  1. Figure Clarity and Accessibility: In Figures 2 and 3, clarify axis labels and significance markers. Ensure each figure panel is labeled in the text clearly, and legends include exact n values and replicate types

We thank the reviewer for this valuable suggestion. We have improved the clarity of Figures 2 and 3 by enlarging the significance markers, refining the axis labels, and ensuring that all figure panels are clearly referenced in the text. In addition, the figure legends now specify the type of replicates used in each experiment.

Reviewer 2 Report

Comments and Suggestions for Authors

Major Comments

RNA-seq data accessibility – The manuscript lacks a mention of where RNA-seq data will be deposited (GEO, ArrayExpress). This is essential for reproducibility.

Replicate clarification – Indicate clearly whether “two independent experiments” refers to biological replicates. For functional assays (migration, ROS, adhesion), include n and specify statistical tests (two-way ANOVA or t-test) in each figure legend.

Figure integrity – Provide uncropped Western blot images in supplementary material with molecular weight markers visible (for TFAM and β-actin). The current supplementary images show different exposure times without ladder reference.

Statistical reporting – Include exact p-values or ranges instead of asterisks only. Define whether data distribution was tested for normality before ANOVA.

Overexpression inefficiency in A172 – The authors state overexpression “likely activated cell death pathways.” Please support this with data (apoptosis markers, cell viability curves) or remove speculative phrasing.

Functional significance of TRAF2/LOXL2 – The discussion implies these are therapeutic targets. The data are correlative; consider softening the language (“potential mediators” rather than “therapeutic targets”) unless functional validation is provided.

  • Line 84–92: Consider summarizing previous findings in fewer sentences for readability.

  • Define abbreviations at first use (e.g., CRISPRa, DCFDA, etc.).

  • In Figure 4C, clarify whether SOD1/SOD2 data were newly generated or from a previous publication.

  • Correct small typos (“siTFAM” appears inconsistently capitalized in text and figure legends).

  • Please ensure all figures follow IJMS graphical guidelines (minimum 300 dpi, consistent labeling).

Author Response

We sincerely thank the reviewers and the editor for their constructive feedback, which has improved the clarity and rigor of the manuscript. 
Below we provide point-by-point responses.

Reviewer #2

Comments and Suggestions for Authors:

Major Comments

  • RNA-seq data accessibility – The manuscript lacks a mention of where RNA-seq data will be deposited (GEO, ArrayExpress). This is essential for reproducibility.

We thank the reviewer for this helpful suggestion. We have prepared all the data for submission to the GEO repository. We attempted to upload the data; however, the NCBI website repeatedly displayed the following error message. As soon as GEO accepts our submission, we will upload all RNA-seq data and provide the corresponding accession ID to the journal and the reviewers.

  • Replicate clarification – Indicate clearly whether “two independent experiments” refers to biological replicates. For functional assays (migration, ROS, adhesion), include n and specify statistical tests (two-way ANOVA or t-test) in each figure legend.

We included the replicates information in each figure legend. The statistical tests are described in results sections, to ensure that the figure captions are not excessively lengthy.

  • Figure integrity – Provide uncropped Western blot images in supplementary material with molecular weight markers visible (for TFAM and β-actin). The current supplementary images show different exposure times without ladder reference.

We thank the reviewer for this valuable comment regarding the presentation of Western blot data. In response, we have included a new supplementary figure showing the uncropped Western blot membranes for TFAM and β-actin, with molecular weight markers indicated.

Although the visible marker bands are partially obscured due to limitations in our equipment’s white light source, we have clearly denoted the corresponding molecular weights with red arrows to aid interpretation. These additions ensure transparency and allow verification of band size and integrity across all blots.

  • Statistical reporting – Include exact p-values or ranges instead of asterisks only. Define whether data distribution was tested for normality before ANOVA.

We thank the reviewer for this helpful suggestion regarding statistical reporting. To improve clarity while maintaining figure readability, we retained the asterisks in the figures to indicate statistical significance and have included the corresponding exact p-values in the figure legends and Results section for each experiment. Additionally, all revised figures now include p-values within the graphs in the supplementary material for full transparency.

Normality of data distribution was assessed for all functional experiments prior to applying parametric tests (ANOVA), ensuring the validity of the statistical analyses.

  • Overexpression inefficiency in A172 – The authors state overexpression “likely activated cell death pathways.” Please support this with data (apoptosis markers, cell viability curves) or remove speculative phrasing.

We appreciate the reviewer’s insightful comment and have revised the text accordingly to remove speculative interpretation. The statement has been rephrased as follows: “By contrast, the A172 cell line exhibited inefficient TFAM overexpression at both multiplicities of infection (MOIs) tested, resulting in a complete loss of viable cells and precluding further functional analyses” in line 316. This revision accurately reflects our experimental observation without implying a specific mechanism that was not directly demonstrated.

  • Functional significance of TRAF2/LOXL2 – The discussion implies these are therapeutic targets. The data are correlative; consider softening the language (“potential mediators” rather than “therapeutic targets”) unless functional validation is provided.

We thank the reviewer for this valuable comment and fully agree that our wording should reflect the correlative nature of the findings. Accordingly, we have softened the language in the revised text to avoid implying direct therapeutic validation. The sentence now reads: “The availability of promising TRAF2-targeting agents, such as liquidambaric acid [61] and anti-TNFR2 antibodies currently under clinical evaluation (NCT04752826) [62], strengthens the rationale for TRAF2 as a potential mediator in TFAM-low GBM.” This revision (line 419) clarifies that TRAF2 is discussed as a potential mediator rather than a validated therapeutic target.

  • Line 84–92: Consider summarizing previous findings in fewer sentences for readability.

We thank the reviewer for this helpful suggestion. In response, we have shortened the text in lines 87–92 to improve readability and conciseness. The revised version now reads:

“We previously reported that TFAM expression declines with increasing tumor grade, with low-grade astrocytomas (AG2 and AG3) exhibiting higher levels than GBM (AG4). Elevated TFAM expression correlated with improved patient prognosis, suggesting a protective role in glioma progression [18]. These findings underscore the close interplay between metabolic reprogramming and malignancy, yet the signaling pathways underlying TFAM dysfunction remain largely undefined.”

This revision effectively summarizes the previous findings in fewer, more cohesive sentences, as recommended.

  • Define abbreviations at first use (e.g., CRISPRa, DCFDA, etc.).

We thank the reviewer for this helpful comment. We have carefully reviewed the entire manuscript and ensured that all abbreviations, including CRISPRa, DCFDA, and others, are defined at their first appearance.

  • In Figure 4C, clarify whether SOD1/SOD2 data were newly generated or from a previous publication.

We thank the reviewer for this important clarification. The SOD1 and SOD2 expression data shown in Figure 4C were obtained from transcriptomic analyses of untreated cells generated concurrently with the NTC and siTFAM samples. These data were used to illustrate the basal antioxidant profiles of the cell lines employed in this study. We have rephrased lines 204-206 in the revised manuscript to improve clarity accordingly.

  • Correct small typos (“siTFAM” appears inconsistently capitalized in text and figure legends).

We thank the reviewer for noting this oversight. We have carefully reviewed the entire manuscript, including figure legends, and corrected all instances of inconsistent capitalization of “siTFAM” for uniformity throughout.

  • Please ensure all figures follow IJMS graphical guidelines (minimum 300 dpi, consistent labeling).

We confirm that all figures have been revised to comply with the IJMS graphical requirements, including a minimum resolution of 300 dpi and consistent labeling across all panels and figure legends. In fact, all images were exported at 600 dpi from Adobe Illustrator to ensure optimal quality.

Round 2

Reviewer 1 Report

Comments and Suggestions for Authors

The authors satisfactorily addressed the previous critique. The manuscript is improved.